# Neuroplasticity-Based Approaches to Sensory Processing Alterations in Autism Spectrum Disorder

**DOI:** 10.3390/ijms26157102

**Published:** 2025-07-23

**Authors:** Maria Suprunowicz, Julia Bogucka, Natalia Szczerbińska, Stefan Modzelewski, Aleksandra Julia Oracz, Beata Konarzewska, Napoleon Waszkiewicz

**Affiliations:** Department of Psychiatry, Medical University of Bialystok, pl. Wołodyjowskiego 2, 15-272 Białystok, Poland; maria.suprunowicz@sd.umb.edu.pl (M.S.); 41207@student.umb.edu.pl (J.B.); 35984@student.umb.edu.pl (N.S.); stefan.modzelewski@sd.umb.edu.pl (S.M.); aleksandra.oracz@sd.umb.edu.pl (A.J.O.); beata.konarzewska@umb.edu.pl (B.K.)

**Keywords:** sensory integration, autism, ASD, neuroplasticity, therapy, sensory alterations

## Abstract

Sensory dysregulation represents a core challenge in autism spectrum disorder (ASD), affecting perception, behavior, and adaptive functioning. The brain’s ability to reorganize, known as neuroplasticity, serves as the basic principle for therapeutic interventions targeting these deficits. Neuroanatomical mechanisms include altered connectivity in the sensory and visual cortices, as well as in the limbic system and amygdala, while imbalances of neurotransmitters, in particular glutamate and gamma-aminobutyric acid (GABA), contribute to atypical sensory processing. Traditional therapies used in sensory integration are based on the principles of neuroplasticity. Increasingly, new treatments use this knowledge, and modern therapies such as neurofeedback, transcranial stimulation, and immersive virtual environments are promising in modulating neuronal circuits. However, further research is needed to optimize interventions and confirm long-term effectiveness. This review discusses the role of neuroplasticity in the etiopathogenesis of sensory integration deficits in autism spectrum disorder. The neuroanatomical and neurotransmitter basis of impaired perception of sensory stimuli is considered, and traditional and recent therapies for sensory integration are discussed.

## 1. Introduction

Autism spectrum disorder is a neurodevelopmental disorder characterized by abnormalities in social and communication skills and a tendency towards limited and repetitive behavior patterns. These traits typically emerge during early childhood and vary widely in severity and presentation [1]. Recent estimates from the Centers for Disease Control indicate that ASD affects approximately 1 in 44 children, with a markedly higher prevalence in boys than in girls [2,3]. However, the observed sex differences in autism spectrum disorder may largely result from diagnostic gender bias. While ASD is often reported as being four times more common in males than females, meta-analyses indicate the true male-to-female ratio is closer to 3:1, suggesting that many autistic girls remain undiagnosed due to subtler symptom presentation [4]. Similarly, the prevalence of ASD in adults is estimated to be around 2.2% [5]. The underlying causes of autism are complex and multifactorial, involving an interplay between genetic susceptibility and early environmental factors such as imbalances in nutrition, exposure to viruses during pregnancy, an impaired immune system, and alterations in gut microbiota [6]. Numerous gene variants have been linked to increased risk, although no single cause has been identified [5].

In addition to the main symptoms of ASD, sensory integration disruptions are also mentioned, which include excessive or insufficient reactions to auditory, visual, tactile, proprioceptive, or other stimuli [7]. This phenomenon can be attributed to alterations in the brain structures implicated in the processes of perception, analysis, and integration of sensory stimuli, such as the cerebellum, sensory cortex, and thalamus [8]. Evidence has also been found of abnormal neurotransmitter secretion in individuals diagnosed with ASD. This abnormality includes disruptions in the release of glutamate (Glu) or gamma-aminobutyric acid (GABA) [9]. Not only altered brain anatomy but also abnormal neuroplasticity may play a role in impaired sensory integration. Neuroplasticity is a complex process that enables the central nervous system (CNS) to adapt structurally and functionally to experiences, maturation, and recovery from injury [10]. This includes genetic, molecular, and cellular mechanisms that modulate synaptic connections and neuronal circuits, resulting in the strengthening or loss of behavior and function [11]. However, neuroplasticity can also result in maladaptive outcomes depending on the stage of neurodevelopment, the extent of neuropathogenic factors, and the integrity of homeostatic regulatory mechanisms [12]. Research indicates that children with predominant sensory hypersensitivity tend to display increased anxiety symptoms, whereas those characterized by sensory hyposensitivity or a strong need for sensory stimulation are more likely to exhibit symptoms associated with attention deficit hyperactivity disorder (ADHD) [13]. Notably, emotional regulation difficulties appear to be elevated across all children with atypical sensory processing, regardless of the specific sensory profile [13]. Given the growing recognition of sensory integration deficits and altered neuroplasticity in ASD, further investigation into their underlying mechanisms is crucial for improving individualized care and intervention approaches.

## 2. Materials and Methods

A search was performed in the following databases: PubMed, Web of Science, and Scopus, using combinations of the following search terms: “autism, autism spectrum disorder (ASD), sensory disturbances, sensory integration, therapy, neuroplasticity”. The literature search and the manuscript writing were performed between November 2024 and July 2025. Since this work is a narrative review, no strict inclusion criteria were established for the selection of articles. During the screening process, studies addressing neuroplasticity, sensory integration, and therapies in ASD were included. Based on titles and abstracts, conference abstracts were excluded. Studies not written in English were also omitted. Additionally, supporting literature on potential therapeutic approaches was included. The suitability for inclusion was evaluated on the basis of the full publication. All reference lists of found articles were screened for usefulness. Mendeley Reference Manager Version 2.131.0 was used to remove duplicates that appeared due to overlapping search terms.

## 3. Neuroplasticity in ASD

Neuroplasticity is defined as the brain’s capacity to establish new neural connections or eliminate superfluous ones in response to environmental stimuli [10]. The processes responsible for this phenomenon are neurogenesis and synaptogenesis. Neurogenesis is defined as the formation of new synaptic connections and neural networks, while synaptogenesis is the elimination of unnecessary synapses, otherwise known as synaptic pruning [14,15]. Neuroplasticity plays a key role in the effective processing of sensory stimuli and in generating adequate responses to these stimuli [16]. For the sensory integration process to work properly, the brain must identify sensory patterns, anticipate their meaning, and create appropriate adaptive responses [16]. It is evident that neuroplasticity is a pivotal factor in the development of these abilities during childhood, with the potential for modification into adulthood [16]. However, in ASD, the plastic reorganization process may be altered, resulting in dysfunctional sensory integration. The inability to adequately filter, prioritize, or adapt to sensory input could stem from atypical synaptic plasticity mechanisms operating during critical developmental windows. The presence of neuroplasticity abnormalities may offer a potential explanation for the etiopathogenesis of autism spectrum disorder. This can involve opening and closing critical periods at suboptimal times, consequently leading to improper organization of individual neural circuits [17]. Another theory postulates that homeostatic plasticity in ASD is maladaptive and results in the destabilization of network activity, such as an imbalance between excitatory and inhibitory responses [18]. Although excitatory–inhibitory neurotransmission disruptions currently play the most important role in the impaired neuroplasticity of ASD, further research is still needed in this area [19].

The role of glutamate (Glu) in neuroplasticity is a subject of considerable interest. Research has indicated that children diagnosed with autism spectrum disorder exhibit elevated plasma levels of this amino acid [20]. Excess glutamatergic activity may contribute to excessive synaptic strengthening or impaired synaptic pruning, potentially resulting in hyperconnected but inefficient networks—a pattern that may underlie atypical sensory processing in ASD. The elements that are subject to disruption in neuroplasticity include abnormalities in long-term potentiation (LTP) and long-term depression (LTD), disruptions in protein synthesis, changes in the morphology and synthesis of the dendritic spine, and abnormalities in synaptic pruning [21]. LTP is a neuroplasticity process that results in a permanent increase in the strength of connections between neurons [21]. It occurs as a result of intense or multiple activation of synapses, which triggers a series of molecular and biochemical mechanisms leading to more efficient signal transmission [21]. A key element of LTP is the activation of glutamate receptors such as N-methyl-D-aspartate receptors (NMDAR) [22]. In the context of ASD, exaggerated or dysregulated LTP may lead to heightened sensory encoding without corresponding regulation, contributing to hypersensitivity and an inability to suppress irrelevant stimuli. LTD, on the other hand, is the opposite of LTP and is responsible for the long-term reduction in synaptic connection strength [21]. This form of neuroplasticity enables the reorganization of neuronal networks and their adaptation to changing conditions [21]. LTD occurs in response to specific patterns of synaptic activity that are usually less intense than those needed to induce LTP [23]. During this process, the number of functioning glutamate receptors, mainly α-amino-3-hydroxy-5-methyl-4-isoxazolepropionic acid (AMPA), in the postsynaptic membrane is reduced, which leads to a weakening of signal transmission [23]. This mechanism enables the brain to modify stored memory traces, eliminate superfluous information, and extinguish reactions to stimuli that have become meaningless [24]. (Figure 1). If LTD processes are impaired in ASD, the brain may fail to properly downregulate responses to previously neutral or habituated stimuli—contributing to persistent hyperreactivity or fixation on repetitive sensory input. Thus, the combined disruption of LTP and LTD mechanisms may not only alter memory and learning but also underlie the fundamental difficulties in adaptive sensory modulation observed in ASD. These neurobiological disruptions offer a compelling link between molecular mechanisms of plasticity and the behavioral phenotypes of sensory integration dysfunction.

It is postulated that an imbalance between excitatory and inhibitory responses is a contributing factor in the etiology of ASD [25]. In mouse models with TSC2 gene mutations—associated with tuberous sclerosis—LTP is enhanced, while LTD is reduced, suggesting that altered synaptic plasticity dynamics may underlie ASD-like behaviors [26]. Similarly, changes in LTD have also been observed in mice with absent FMR1 protein, which corresponds to the fragile X syndrome, and in mice with Angelman syndrome [19,27]. The loss of FMR1 has been linked to an increase in protein synthesis due to increased LTD [28]. This phenomenon has been observed in conjunction with the overproduction of immature dendritic spines, which may be a consequence of the aforementioned factors [28,29]. Furthermore, the loss of the TSC1 or TSC2 genes, in addition to affecting LTD, leads to the hyperactivation of the mammalian target of rapamycin complex 1 (mTORC1) protein complex, which is a component of the mechanistic target of rapamycin (mTOR) pathway [29]. This pathway controls mRNA translation, thus regulating protein synthesis [29]. Hyperactivation of the mTOR pathway has been linked to a deficiency in dendritic spine pruning, which occurs as a result of the loss of mTOR-dependent autophagy [30]. A post-mortem examination of the brain tissue of individuals diagnosed with ASD revealed a higher density of dendritic spines in their cerebral cortex neurons. Furthermore, it was observed that the synaptic pruning process is less effective as a result of impaired microglial activation [26]. These findings suggest a failure to eliminate redundant synaptic connections during critical developmental windows, which may contribute to atypical sensory processing. Finally, mutations in the *SHANK* gene family, which encode postsynaptic scaffolding proteins, have been identified in ASD models and are known to impair the formation and maturation of dendritic spines [31,32]. As *SHANK* proteins are central to the structural and functional integrity of excitatory synapses, their dysfunction may exacerbate network instability and disrupt neuroplasticity mechanisms essential for sensory integration.

Several other genes critical for synaptic development and neuroplasticity are implicated in ASD. One of the most extensively studied is the gene encoding brain-derived neurotrophic factor (BDNF), a neurotrophin essential for synaptic growth, neuronal survival, and the modulation of long-term potentiation [33]. Altered BDNF levels have been reported in individuals with ASD and are thought to affect synaptic pruning and the maturation of neural circuits, potentially contributing to abnormal connectivity patterns observed in the disorder [34,35]. Another key gene is *MECP2*, which encodes methyl-CpG-binding protein 2, a transcriptional regulator involved in activity-dependent gene expression [36]. While *MECP2* mutations are classically associated with Rett syndrome, its role in regulating synaptic plasticity—particularly through the modulation of BDNF expression—makes it relevant to the pathophysiology of ASD as well [37]. *MECP2* dysregulation can disrupt neuronal maturation and spine morphology, leading to impaired cortical circuitry [38]. The gene synaptosomal-associated protein 25 (*SNAP-25*) encodes a presynaptic protein involved in the SNARE complex and is essential for neurotransmitter release and synaptic vesicle fusion [39]. Variants in *SNAP-25* have been linked to ASD and other neurodevelopmental disorders, suggesting a shared vulnerability rooted in synaptic dysregulation [40]. Disruptions in *SNAP-25* function may impair synaptic efficiency and plasticity, particularly in circuits involved in attention, executive functioning, and social behavior [41]. Together, these genes highlight that the molecular disruptions contributing to ASD are not limited to a single synaptic mechanism but encompass a wide network of proteins responsible for the dynamic regulation of neuroplastic processes throughout development. Their interactions with signaling pathways, as well as their impact on neurotransmitter balance, underscore the multifactorial basis of altered neurodevelopment in autism.

Collectively, these molecular and cellular alterations suggest that disrupted neuroplasticity—manifested through impaired synaptic pruning, altered protein synthesis, and aberrant dendritic spine development—may underlie the atypical sensory behaviors observed in ASD. By destabilizing the balance between excitation and inhibition, these mechanisms interfere with the brain’s ability to adaptively process sensory input, supporting the hypothesis that neuroplasticity deficits are core contributors to sensory integration dysfunction in ASD.

## 4. Sensory Disturbances and Changes in Brain Anatomy in ASD

### 4.1. Disturbances in Auditory Stimulus Processing

A common symptom in patients with autism spectrum disorder is hypersensitivity to sound [42]. Individuals diagnosed with ASD have been observed to exhibit disproportionate emotional responses to a range of auditory stimuli, both quiet and loud [43]. This condition has been referred to in the literature as “misophonia” or “phonophobia”, defined as an aversion to sound [43]. Lucker and Doman posit a hypothesis that the hypersensitivity to sounds exhibited by children diagnosed with ASD does not stem directly from dysfunction within the auditory system [44]. Rather, they contend that this hypersensitivity is underpinned by emotional factors [44]. The authors contend that non-classical auditory pathways and their connections to the limbic system, which is responsible for emotions, are pivotal in this context [44]. They emphasize that the connection between the auditory and limbic systems is located deep in the temporal lobe of the brain [44]. The vagus nerve, which is one of the most important cranial nerves, may also play a role in the body’s response to sound and has a strong connection to the limbic system. In the context of auditory hypersensitivity, the vagus nerve amplifies these reactions, causing some people to experience severe discomfort and anxiety in response to certain sounds [45].

These findings highlight that auditory hypersensitivity in ASD may stem from altered cross-talk between sensory and emotional processing circuits, rather than solely from primary auditory dysfunction. It suggests that neuroplasticity changes affecting limbic–auditory integration may underlie heightened auditory reactivity and emotional dysregulation in ASD.

One strategy employed by children with ASD to cope with overwhelming auditory stimuli is echolalia, defined as the repetition of words and sentences. In some cases, this behavior may be employed as a means of regulating their own responses to an excessively stimulative auditory environment [46]. It can be interpreted as a compensatory behavioral adaptation in response to impaired sensory filtering mechanisms, further supporting the role of altered plasticity in auditory–limbic circuitry. Disturbed speech processing and difficulties in focusing on verbal communication in the presence of background noise are also consequences of auditory hypersensitivity [47]. Research has demonstrated that some children diagnosed with ASD encounter challenges in comprehending and producing speech [48]. These challenges subsequently result in delays in the development of their vocabulary and the capacity to formulate coherent sentences [48]. Speech–language delay may reflect dysfunction in auditory processing regions that fail to modulate input effectively, potentially due to reduced plasticity during critical developmental windows.

### 4.2. Hypersensitivity to Light

Photophobia or hypersensitivity to light is a common sensory phenomenon observed in autism [49]. The visual cortex is the main area of the brain responsible for analyzing visual information [50]. Individuals with autism spectrum disorder may have altered activity in this structure, which can result in distorted perception of visual stimuli, including light [50]. Instead of selectively processing signals, which allows a person to ignore the background and focus on important information, the brain of a person with ASD may react to all stimuli with greater intensity [50]. Patients may experience problems, such as difficulty distinguishing relevant stimuli from the background, excessive focusing on object details, fixations on visual impressions, and excessive visual stimulation from incoming light [51]. Generalized hyperresponsiveness suggests a lack of proper synaptic filtering, likely driven by alterations in neuroplastic mechanisms that support sensory habituation and salience attribution [51]. A functional magnetic resonance imaging (fMRI) study showed that people on the autism spectrum exhibit increased activity in the posterior regions of the brain, including the primary cortex (V1) and the extrastriate cortex, while showing decreased activity in the frontal regions [52]. In the inaugural fMRI study using electroencephalography (EEG), it was observed that the ventral occipitotemporal regions showed increased activation [52,53]. These brain regions are an integral part of early visual processing and are involved in the processing of visual object imagery [52]. Other studies have shown that individuals with ASD exhibit hyperactivation of the occipital regions during visual detection tasks [44,53]. Abnormal patterns of brain activity may reflect impaired synaptic pruning or faulty top-down modulation, both of which are functions closely linked to experience-dependent plasticity. It may contribute to the enhancement of certain stimulus characteristics, such as increased sensitivity to light [54].

Such findings reinforce the hypothesis that altered neuroplasticity may underlie visual hypersensitivity in ASD by failing to refine and stabilize visual networks during development. Disturbances in visual cortical tuning may lead to not only to photophobia but also to broader sensory integration deficits.

### 4.3. Impaired Processing of Tactile Stimuli

ASD patients may show a reduced response to tactile stimuli, but some also experience tactile hypersensitivity [55,56]. Studies have shown that high-functioning children with ASD can react very strongly to touch, perceiving it as unpleasant or uncomfortable, and they also show hypersensitivity to proprioceptive stimuli and pain [57]. However, it is not evident that they will encounter challenges with stereognosis, defined as the ability to discern tactile characteristics such as shape or texture [7]. Therefore, an aberration in the processing of tactile stimuli results in challenges when it comes to the manipulation of objects [58]. A multitude of studies have demonstrated that individuals diagnosed with ASD frequently exhibit atypical sensitivity to touch and an incapacity to acclimate to repeated stimuli [58]. A potential cause is considered to be changes in GABAergic feedback loops, which can contribute to abnormal tactile sensitivity [59]. Children with ASD who are hypersensitive to stimuli have insufficient GABA inhibition, which results in impaired processing of low-intensity stimuli [59]. Alterations in inhibitory neurotransmission suggest disrupted sensory gating mechanisms, which are essential for habituation to repeated tactile input and depend on experience-driven plasticity in somatosensory circuits. Deficient GABAergic inhibition may therefore reflect altered developmental plasticity in ASD, leading to both hyperreactivity and dysregulation of sensory integration. Children who demonstrate deficiencies in non-verbal communication exhibit a reduction in tactile sensitivity [60]. The neuronal and behavioral effects of oxytocin, a hormone responsible for the creation of social bonds and also released by tactile stimuli, are weaker in people with ASD, which means that children with autism are less likely to seek tactile contact in human interactions [61].

Diminished oxytocinergic response may further contribute to the social avoidance of touch, reinforcing atypical tactile development and weakening the bidirectional feedback loops between sensory processing and social behavior. The reduced plasticity of these feedback circuits may underlie long-term alterations in both sensory and emotional responses to touch.

### 4.4. Incorrect Texture Processing

A significant number of children with autism have a preference or bias towards certain food textures [62]. It has been observed that more than half of children with ASD have a food selectivity characterized by a limited range of food products they eat [62]. An autistic person may prefer meals that are easy to swallow, with a specific texture, consistency, color, or taste, from a specific plate and using specific utensils, but not those that require significant chewing time [63]. It has been observed that children who were exposed to intense or stressful stimuli in early childhood, such as uncomfortable clothing, difficult environmental conditions, or negative experiences with certain textures, may develop hypersensitivity as a protective mechanism against further exposure to such unpleasant stimuli [49]. These observations support the idea that sensory hypersensitivity may be reinforced by maladaptive neuroplastic changes, in which negative tactile experiences strengthen avoidance circuits and inhibit sensory habituation. It may explain the rigidity and resistance to sensory novelty often seen in ASD. Individuals with heightened sensory sensitivity frequently exhibit challenges in proprioception, defined as the awareness of one’s own body in space [64]. In the event of an impaired proprioceptive sense, bodily reactions to tactile stimuli may be amplified, leading to heightened sensitivity to various textures [64]. The sensory cortex, situated within the parietal lobe of the brain, is the region responsible for receiving and processing tactile stimuli [62]. In typical cases, the brain analyzes tactile stimuli, filtering and normalizing them so that the touching of textures, such as a rough surface or soft material, is perceived according to its intensity and character [62]. In individuals with heightened sensitivity to textures, the sensory cortex may exhibit increased reactivity, leading to the interpretation of stimuli as more intense or irritating than they actually are. This heightened sensitivity to textures may be accompanied by impaired communication between the sensory cortex and other brain regions, including the limbic system [43]. In addition, neurotransmitters play an important role in the regulation of sensory signals. In cases of increased sensitivity to textures, their imbalance may be observed, leading to an inability of the nervous system to adequately modulate responses to stimuli. Research has demonstrated that the decreased level of the inhibitory neurotransmitter GABA can result in sensory signals being perceived more intensely due to the absence of a mechanism to dampen their intensity [65]. Aforementioned neurotransmitter imbalance points to a broader dysregulation of sensory plasticity, where excitatory–inhibitory imbalance leads to inefficient neural adaptation and excessive salience attribution to non-threatening tactile stimuli.

### 4.5. Inadequate Pain Response

Incorrect processing of tactile stimuli can also manifest as an elevated pain threshold, which can result in frequent injuries that are disregarded by the child [66]. Some individuals with autism may only react to stronger pain stimuli [67]. The phenomenon of hypersensitivity to pain in ASD can be understood as the result of a complex interaction between neurological, sensory, and psychological factors [7]. The amygdala, which plays a role in emotional processes and pain reactions, can be either overactive or underactive in individuals with autism [68]. In contrast, the insula, which is responsible for the subjective perception of pain, may show reduced activity, thus reducing the perception of pain reaction [69]. Such dysregulation in the limbic and interoceptive circuits may result in mismatched affective and sensory processing of painful stimuli. It suggests that altered plasticity in these regions contributes to the atypical calibration of pain thresholds in ASD. Neurotransmitters such as serotonin, endorphins, and dopamine, which play a role in the pain perception process, may also be relevant in its disturbed perception [59,70]. Endorphins, in particular, have been shown to reduce the pain response and, over time, can cause pleasant sensations for stimuli perceived as unpleasant by neurotypical people [59,71]. Increased endorphin levels are observed in children with ASD who engage in self-biting, suggesting that they may experience a lack of pain perception and even derive pleasure from this act. This is thought to be due to the activation of the reward center in the brain [71]. The findings illustrate that pain-related behaviors in ASD may reflect altered homeostatic plasticity within reward and nociceptive networks. Understanding these changes offers insight into why pain expression in ASD can diverge significantly from typical developmental patterns.

The complexity of sensory integration disorders emphasizes the heterogeneity of autism spectrum disorders. Selected elements of the basis of these disruptions are presented in Table 1.

## 5. Conventional Sensory Therapies Based on Neuroplasticity

### 5.1. Ayers Sensory Integration Therapy

Ayres Sensory Integration Therapy (ASI) is a recognized therapeutic approach pioneered by Jean Ayres, originally designed to support children facing learning and behavioral challenges, and the intervention is based on research into neuroplasticity, emphasizing the potential of the nervous system to change [72]. The ASI approach suggests that active engagement in individually tailored sensorimotor activities, contextualized in play, promotes adaptive behaviors through the neuroplastic changes that occur in response to these experiences [72]. The notion that the brain is capable of adapting to experience is a concept that has garnered support from the findings of neuroscience research. Specifically, studies demonstrate that experience-dependent learning contributes to the shaping of both behavior and brain function. A number of studies have demonstrated that rodents in an enriched environment, where they are permitted to explore and interact with toys, exhibit alterations in brain structure and organization [73,74]. These findings further substantiate the notion of neuroplasticity, underscoring the capacity of the brain to form new synaptic connections throughout life in response to environmental enrichment [73]. Participation in an enriched environment with new sensory, motor, and cognitive challenges can induce lasting changes in brain function [74]. The researchers concluded that several key principles of environmental enrichment are consistent with the goals of ASI intervention, suggesting that ASI can effectively facilitate changes in the brain through targeted sensory and motor experiences [74]. The results indicate that children who received the ASI intervention scored higher on the Goal Attainment Scales (GASs) for participation in daily activities and functional skills and showed significantly greater improvement on the Pediatric Disability Inventory for self-care and socialization, compared to the control group [75,76]. This improvement was achieved through a structured treatment protocol, ensuring fidelity through a manual-based approach and a validated ASI fidelity measure [75,76]. Another study showed that implementing Ayres Sensory Integration Therapy twice a week for 60 min over a period of 12 weeks can improve positive behaviors, especially in the areas of communication (both expressive and receptive), socialization (including coping skills), and daily living skills (including personal and social life tasks) [77].

Neurobiologically, ASI is based on the principle of experience-dependent neuroplasticity, involving systems that integrate proprioceptive, vestibular, and somatosensory signals within the sensory cortex, cerebellum, and frontoparietal connections [72]. By providing repetitive, salient sensory experiences, ASI fosters cortical reorganization in circuits underlying sensory modulation, attention, and motor planning—domains consistently impaired in children with ASD [72]. These findings underscore the value of ASI not only as a behavioral intervention but as a biologically informed modality capable of addressing core neurodevelopmental dysfunctions.

### 5.2. Sensory Integration Therapy

Sensory integration therapy (SIT) is a therapy based on the theory that adaptive behavior is influenced by the relationship between behavior and neurological processes in the central nervous system [78]. It has been hypothesized that this therapeutic modality facilitates the perception of a variety of sensory stimuli, encompassing visual, auditory, tactile, proprioceptive, and vestibular stimuli [78]. This, in turn, assists the child in engaging with their surroundings in an effective manner. Over time, aforementioned process supports healthy cognitive, motor, behavioral and emotional development [78]. To date, the effectiveness of SIT has been controversial [79,80]. The main reason for the ongoing controversy is the inconsistency in how sensory integration therapy is applied in each study. SIT has proven effective in several conditions, such as cerebral palsy, ASD, ADHD, intellectual disability, and developmental disorders [78]. However, the effect size was greatest for cerebral palsy, followed by autistic disorders, and then ADHD [79]. The most effective therapy for ASD, according to the results of a meta-analysis published in the World Journal of Clinical Cases, is 1:1 therapy with a therapist or a therapy session lasting 40 min [78]. A greater effect was observed in the areas of social skills, adaptive behavior, and sensory processing functions [78].

On a neurobiological level, SIT is thought to facilitate synaptic reorganization in cortical and subcortical structures involved in multisensory processing, particularly the parietal cortex, cerebellum, and thalamus [81]. It engages the brain’s ability to modulate sensory input through repeated and structured exposure, potentially enhancing functional connectivity within sensory networks. Evidence from functional MRI and EEG studies suggests that sensory integration interventions can reduce overactivation in sensory cortices and improve thalamocortical regulation, which is frequently altered in individuals with ASD [82,83]. These neuroplastic adaptations may underlie observed improvements in attention, motor planning, and behavioral regulation.

### 5.3. Snoezelen Therapy

Snoezelen therapy, also called controlled multisensory environment (MSE), takes place in a sensory room that has been designed to stimulate multiple senses through the use of lighting effects, colors, sounds, music, scents, and other sensory stimuli [84]. It is believed that MSE improves cognitive abilities and aids learning, while encouraging eye contact, shared attention, shared enjoyment, and better communication, all of which help to reduce restrictive and repetitive behaviors [84]. This allows the child to experience and manage their sensory reactions, gradually increasing their ability to process and integrate sensory information from the environment. The few studies conducted to date on interventions in ASD have shown improvements in sustained attention, developmental skills and challenging behaviors [85]. Although the mechanisms of action are not yet fully understood, Snoezelen therapy is believed to influence the limbic system and HPA axis regulation by reducing stress levels and improving affective states [86].

### 5.4. Animal-Assisted Intervention

Animal-assisted intervention (AAI) involves incorporating animals such as dogs, horses, dolphins, rabbits, guinea pigs, and llamas into the therapeutic process as part of autism treatment [87]. It is one of the most promising therapies aimed at repairing the underlying impairments in children with ASD [87,88,89]. Moreover, researchers observed that the rhythmic movement of horse riding can specifically stimulate the vestibular system in children with ASD, which can improve speech production and support better learning outcomes [89,90]. It may also potentially influence sensory–motor integration in the cerebellum and brain stem. At the same time, riders have to actively manage their body movements, which strengthens their ability to exercise voluntary control and improves non-verbal communication skills [89]. Another meta-analysis on nature-based interventions (NBIs) showed a positive relationship between horse-assisted therapy and goal attainment, as well as between nature-based therapy and parent–child relationships [91]. Furthermore, experimental learning has been indicated as a way to improve the short-term sensory and behavioral outcomes in children with ASD [91]. What is important is that contact with animals may activate the reward system, which can promote social engagement and motivation.

### 5.5. Music Therapy

Several studies have shown that music therapy (MT) can effectively improve the social skills of children with ASD [92]. Individuals with ASD have been observed to demonstrate activation in the cortical and subcortical regions of the brain during exposure to both happy and sad music [92,93]. This activation is typically observed to a greater extent than in individuals without ASD, and is particularly pronounced in response to non-musical emotional stimuli [93]. It may be attributed to music’s capacity to engage distributed neural circuits involved in emotion, attention, and communication. Given that individuals with ASD often exhibit atypical processing of affective cues, music may provide an accessible medium through which to rehearse and strengthen these networks in a non-threatening context.

In another review, MT did not show improvement in symptom severity and receptive vocabulary, but significant improvements were observed in brain connectivity, family quality of life, and social communication skills after 8–12 weeks of MT [94]. The results indicate that MT can be effective in increasing social interaction among children with ASD. A comfortable music program can support children in acquiring social skills and adapting to society [94]. However, the number of eligible studies is small, so all conclusions regarding MT as an ASD therapy should be applied with caution.

Neuroimaging and electroencephalogram (EEG) studies reveal that music therapy activates a distributed brain network, including the temporal gyrus, premotor cortex, amygdala, and cerebellum, suggesting that MT engages both emotional and cognitive circuits [95]. In children with ASD, music may modulate brain connectivity between auditory and motor regions, supporting improved coordination and communication [96]. EEG studies have shown that music therapy can increase alpha and theta, and reduce beta activity, often associated with increased relaxation and attentional engagement [97]. Additionally, MT may enhance dopaminergic transmission in reward-related circuits, potentially increasing social motivation and reducing avoidance behaviors in ASD [98].

### 5.6. Parental Involvement in Therapeutic Interventions for ASD

Parental involvement is recognized as a crucial component of effective therapeutic interventions in children with ASD [99]. Studies have shown that interventions involving caregivers—especially parent-mediated therapies—lead to improved outcomes in core ASD symptoms, including social communication and adaptive behavior [99]. The Preschool Autism Communication Trial (PACT) demonstrated that parent-delivered interventions significantly enhanced the quality of parent–child social interaction and sustained improvements in communication skills up to six years post-intervention [100]. Parent-mediated interventions may influence brain plasticity [100]. Increased parental responsiveness has been associated with enhanced functional connectivity in the social brain network, including the prefrontal cortex [101]. Additionally, oxytocinergic signaling, which supports social bonding and affiliative behavior, may be modulated through positive parent–child interactions [101]. Elevated endogenous oxytocin levels following supportive parenting have been observed in both neurotypical and ASD populations, suggesting a biochemical mechanism through which parental involvement might facilitate social learning and stress regulation [102]. Parental involvement is not merely an adjunct to therapy but rather a neurodevelopmentally relevant for comprehensive ASD care. Future research should further explore how tailoring interventions to individual parent–child dyads—taking into account neurobiological markers, stress levels, and family dynamics—may optimize therapeutic outcomes.

## 6. Modern Sensory Integration Therapies

### 6.1. Virtual Reality/Augmented Reality Technologies

Virtual reality (VR) technologies can accurately present sensory stimuli and be integrated with human sensing technologies to automatically detect sensory responses, and thus can improve the objectivity and sensitivity of sensory assessment compared to traditional questionnaire-based methods [103]. Modern therapies using virtual reality technology are finding increasing use in the treatment of psychiatric disorders. Specifically, HMD with motion-capture VR games has proven to be an effective tool in the treatment of pain of various etiologies, and Cave Automatic Virtual Environment (CAVE) has shown efficacy in the treatment of ASD and other neurodevelopmental disorders [103]. Virtual reality-incorporated cognitive behavioral therapy (VR-CBT) had positive effects on sensory and motor functions in autism spectrum disorder [104].

VR enables precise modeling of sensory stimulation, which translates into controlled activation of selected brain regions, such as the somatosensory cortex, insula, and visuomotor and visuo-tactile integration areas [105]. The use of VR in ASD may support the reorganization of neural networks through repetitive exposure and feedback. It not only facilitates individualized and reproducible stimulation of sensory modalities but also harnesses neuroplastic mechanisms to support cortical reorganization. This makes VR a promising adjunct to traditional sensory and cognitive therapies, particularly in individuals who may not respond well to conventional approaches.

### 6.2. Cannabinoids

Cannabinoids modulate synaptic activity through CB1 receptors predominantly located in the brain, including the prefrontal cortex, amygdala, hippocampus, and cerebellum [106]. These receptors modulate signaling pathways, regulating synaptic transmission and plasticity. These regions are implicated in emotional regulation, sensory processing, and social behavior—all areas of dysfunction in ASD [107]. Moreover, cannabinoid signaling may influence glutamatergic and GABAergic transmission, potentially restoring the excitatory-inhibitory balance, which is often disrupted in autism [59,106]. Animal studies suggest that cannabinoids can affect neural development and plasticity, and preliminary human data indicate the modulation of connectivity within the default mode network and limbic system [108]. Despite promising results, further mechanistic and longitudinal studies are needed to confirm these effects.

Additionally, cannabinoids have been shown to play a protective role in cases of neurodegeneration and brain damage [109]. Some studies point to a role for the use of cannabinoids in the treatment of ASD, with treatment effects including a reduction in symptoms of hyperactivity, tantrums and self-injury, sleep disturbances, anxiety, agitation and depression, as well as improvements in cognitive function, sensory sensitivity, attention, social interaction and language [110]. However, randomized trials lack better evidence of such therapeutic intervention.

### 6.3. Bioneurofeedback

The efficacy of EEG-based neurofeedback therapy is predicated on the brain’s neuroplasticity, which is engineered to instigate enduring self-regulation of modified neuronal activity [111]. The practice of bioneurofeedback involves the acquisition of skills that enable individuals to self-regulate their brain’s oscillatory activity, thereby exerting a direct influence on the central nervous system [112]. The utilization of bioneurofeedback in the management of pain and excessive sensory activation has been a subject of interest in recent research [113,114]. Research indicates changes in frontal cortex activity and an increase in life satisfaction after bioneurofeedback in adults with sensory hyperactivation, but at the same time, no significant changes were observed in alpha brainwaves, which are key to assessing the effectiveness of neurofeedback [113]. The researchers suggest that further research be conducted to evaluate other oscillatory bands [113].

Bioneurofeedback directly influences EEG oscillations, teaching patients to modulate activity in specific bands like alpha, or beta, which may affect neuronal excitability and executive functions [115]. These changes may reflect improvements in somatosensory network synchronization. Neurofeedback is a promising, non-invasive method of using neuroplasticity to improve dysfunctional sensory and cognitive circuits. Observed changes in brain activity and clinical symptoms suggest that this approach may be a valuable adjunct to existing therapies, particularly for individuals with heightened sensory reactivity or self-regulation disorders.

### 6.4. Brain Stimulation Techniques

Transcranial magnetic stimulation (TMS) is a non-invasive brain stimulation technique in which a variable magnetic field is used to induce a small electric current within the brain [116]. TMS can be used as an ASD biomarker because most studies indicate increased hyperplasticity in these individuals [117]. The application of non-invasive brain stimulation (NIBS), which utilizes transcranial direct current stimulation (tDCS) or repetitive transcranial magnetic stimulation (rTMS), has been demonstrated to be an effective method of restoring sensory functions in stroke patients [118]. In the context of autism, the efficacy of NIBS methods in addressing repetitive behaviors and enhancing sociability, as well as executive and cognitive functions, has been demonstrated [119].

TMS and tDCS target cortical excitability and plasticity by modulating resting membrane potentials and local field potentials [120]. In individuals with ASD, these techniques have been shown to alter activity in key regions such as the dorsolateral prefrontal cortex and motor, visual, and multidemand network (MDN) region—areas involved in executive function and social cognition [121]. Repetitive TMS (rTMS) may also increase long-range functional connectivity, counteracting the atypical neural patterns observed in ASD [122]. Furthermore, tDCS has been associated with the modulation of GABA and glutamate levels, possibly contributing to improved behavioral flexibility and reduction in repetitive behaviors [123].

A summary of selected therapeutic approaches used in ASD, their reported effects, and limitations is presented in Table 2.

## 7. Conclusions

Sensory dysregulation is a key aspect of autism, affecting the daily functioning of individuals diagnosed with ASD. These deficits include hypersensitivity or insensitivity to sensory stimuli, difficulties in processing information, and impaired motor coordination. They affect adaptability, social interactions, and cognitive processes, which underscores their importance in the diagnosis and treatment of autism. The neurobiological basis in autism includes abnormalities in brain structures responsible for sensory processing, such as the somatosensory cortex, thalamus, and connections between the sensory cortex and other areas of the brain. These dysfunctions are linked to abnormalities in neuroplasticity, the brain’s ability to adapt and reorganize under the influence of experience. In autism, both reduced synaptic plasticity and overcompensation in certain areas of the brain are observed, which can lead to abnormal processing of stimuli.

Therapies based on neuroplasticity play a key role in improving the functioning of people with autism spectrum disorder and abnormal sensory integration. Neuroplasticity, defined as the brain’s capacity for reorganization and adaptation, underlies both conventional therapeutic methods and recent advancements in the field. Therapeutic effects are achieved by Ayers Sensory Integration Therapy, music therapy, Snoezelen therapy, or animal-assisted therapy. Advances in neuroscience have led to the development of new methods, such as neurofeedback training, which uses EEG to regulate brain activity through feedback mechanisms that support self-regulation and attention, and brain stimulation techniques which influence synaptic plasticity and can improve cognitive and behavioral functions. Virtual reality-based therapy engages patients in controlled therapeutic environments to support the learning of social and sensory skills, while interventions based on games and mobile applications are tailored to individual needs, promoting neuroplasticity. A potential treatment option could be the use of cannabinoids, which are also responsible for changes in synaptic transmission and may be involved in neuromodulation.

Despite promising results, there are several limitations, including the lack of standardized criteria for evaluating the effectiveness of many emerging therapies, the limited number of randomized controlled trials, and the need for long-term studies to evaluate the durability of therapeutic effects. The heterogeneity of ASD symptoms complicates the standardization of therapeutic approaches, while individual factors such as age, level of functioning, and comorbidities significantly affect the results of therapy.

## Figures and Tables

**Figure 1 ijms-26-07102-f001:**
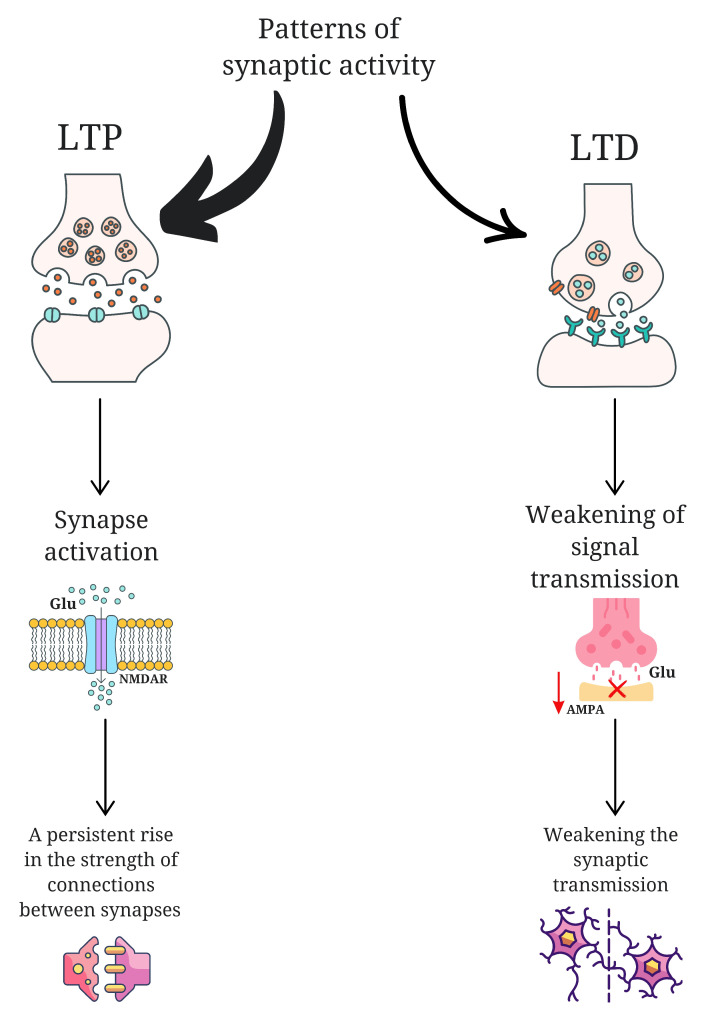
A comparison of the long-term potentiation (LTP) and long-term depression (LTD). LTP acts through glutamate N-methyl-D-aspartate receptors (NMDAR), leading to the activation of synapses and strengthening of connections between neurons. LTD is expressed as less intense patterns of synaptic activity, causing a reduction in α-amino-3-hydroxy-5-methyl-4-isoxazolepropionic acid receptors (AMPA) in the postsynaptic membrane, which results in the weakening of synaptic transmission and a reduction in connections between neurons. This figure was generated using Canva; https://www.canva.com (accessed on 4 March 2025).

**Table 1 ijms-26-07102-t001:** Selected abnormalities potentially responsible for the pathogenesis of sensory integration disorders in autism spectrum disorder (ASD).

Neuroanatomical and Neurotransmitter Abnormalities in Sensory Processing Disorders in ASD	
Incorrect processing of auditory stimuli	Abnormalities in the limbic system [44] Abnormal vagus nerve response [45]
Hypersensitivity to light	Abnormalities in the visual cortex [50] Abnormalities in the primary cortex (V1) and the extrastriate cortex [52] Decreased activity in the frontal regions [52] Hyperactivation of the occipital regions [44,53]
Incorrect processing of tactile stimuli	Insufficient GABA inhibition [59] Abnormal oxytocin level [61]
Incorrect texture processing	Increased reactivity of the sensory cortex [62] Impaired communication between the sensory cortex and other brain regions, including the limbic system [43] Decreased level of the inhibitory neurotransmitter GABA [65]
Inadequate pain response	Overactive or underactive amygdala [68] Insula reduced activity [69] Increased endorphin levels [71]

**Table 2 ijms-26-07102-t002:** Summary of therapies used in autism spectrum disorder targeting sensory processing: therapeutic effects and limitations.

Therapy	Description	Therapeutic Effects	Limitations
Ayers Sensory Integration (ASI)	Therapy based on individualized sensorimotor activities through play [72]	Improvement in self-care, social and communication skills; increased participation in daily activities [74,75,124]	Requires high treatment fidelity; therapist must be trained in ASI; limited availability [16]
Sensory Integration Therapy (SIT)	Based on the theory of sensory processing; targets multiple sensory modalities, including visual, tactile, and vestibular input [78]	Improvements in sensory, social, and adaptive functioning [78]	Inconsistent implementation across studies; efficacy remains controversial [79,80]
Snoezelen Therapy (Multisensory Environment)	Controlled multisensory environment using light, sound, and touch [84]	Reduces challenging behaviors, improves attention, eye contact, and social interaction [85]	Limited number of studies; subjective outcome measures [86]
Animal-Assisted Intervention (AAI)	Use of animals as part of therapy [87]	Enhances cognitive functions, motor control, social engagement; improves child–parent interaction [88,89,91]	High cost; access issues; requires further controlled studies [125]
Parental involvement	Training and involving parents directly in delivering therapeutic activities at home or during sessions [100]	Improved communication, social reciprocity, and generalization of skills across settings [99,100]	Requires high parental commitment; variation in fidelity; outcomes depend on parent–therapist alliance [126]
Music Therapy (MT)	Use of music listening and participation as therapeutic medium [92]	Improves social interaction, brain connectivity, family quality of life [92]	No improvement in core symptoms; few high-quality studies [94]
Virtual/Augmented Reality (VR/AR)	Use of immersive digital environments for sensory stimulation and feedback [103]	Improves motor and sensory functions; objective assessment of responses [104]	Expensive equipment; limited standardization; early-phase research [104]
Cannabinoids	Modulation of neurotransmission via CB1/CB2 receptors [106]	Reduces hyperactivity, aggression, sleep disturbances; improves attention and communication [110]	Lack of randomized trials; potential side effects; preliminary data [110]
Bioneurofeedback (EEG-NF)	Self-regulation of brain oscillatory activity using EEG feedback [111]	Improved frontal cortex activation and quality of life in sensory hyperactivation [113]	No significant change in alpha waves; requires more research into EEG patterns [127]
Non-invasive Brain Stimulation (TMS/tDCS)	Magnetic/electrical stimulation to influence neural activity and plasticity [116]	Improvement in executive function, social behavior, and repetitive behaviors [119]	Still experimental in ASD; needs protocol standardization and more trials [128]

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
