# Peer review of "Neuroplasticity-Based Approaches to Sensory Processing Alterations in Autism Spectrum Disorder"

_ijms, 2025, doi:10.3390/ijms26157102_

Round 1

Reviewer 1 Report

Comments and Suggestions for Authors

This review provides a brief description of the mechanisms associated with ASD and the various treatments available to mitigate symptoms of ASD. The review is well written and would be helpful for readers to gain perspective in this field. However, I have some concerns that need to be addressed:

  1. The introduction is weak. I believe there should be more details included in ASD, such as statistics and etiology.
  2. In many instances, the conclusions drawn from the cited literature are insufficient. There are only sentences stating what has been found, but a proper connection to the discussion needs to be made. And this is throughout the article.
  3. There should be another table listing different therapies used to treat ASD, their outcomes, and associated limitations.

Author Response

Dear Reviewer,

We would like to thank you for your thorough reading of our manuscript and for the valuable comments that helped us improve the quality, clarity, and depth of our work. Below, we provide a point-by-point response to each of the concerns raised:

  1. Comment 1: “The introduction is weak. I believe there should be more details included in ASD, such as statistics and etiology.”
    Response: Thank you for this important comment. We have substantially revised the Introduction section to include more detailed information on the prevalence of ASD. We have also expanded the discussion on etiology to include both genetic and environmental factors, as well as their interactions. These additions are intended to provide a clearer context for the therapeutic challenges addressed in the rest of the review (see revised Introduction, lines 27–42).
  2. Comment 2: “In many instances, the conclusions drawn from the cited literature are insufficient. There are only sentences stating what has been found, but a proper connection to the discussion needs to be made. And this is throughout the article.”
    Response: We appreciate this thoughtful observation. To address this, we have carefully revised multiple sections of the manuscript to deepen the interpretation of cited findings. We now more clearly articulate the implications of key studies, how they support or contrast with other literature, and what they suggest about underlying mechanisms or therapeutic applications. In particular, Sections 3, 4, 5 and 6 have been enriched with additional explanatory context and discussion to strengthen the narrative and coherence between findings and their relevance.
  1. Comment 3: “There should be another table listing different therapies used to treat ASD, their outcomes, and associated limitations.”
    Response: In response, we have added Table 2 (page 15), which summarizes various therapies used or potentially used in ASD, including their descriptions, therapeutic effects, and known limitations.

Once again, we are grateful for the insightful and constructive feedback. These suggestions have helped us improve the manuscript considerably.

Sincerely,

The Authors

Reviewer 2 Report

Comments and Suggestions for Authors

In this narrative review, the authors discuss the role of neuroplasticity in the etiopathogenesis of sensory integration deficits in autism spectrum disorder (ASD). The neuroanatomical and neurotransmitter bases of impaired sensory perception are considered, along with both traditional and emerging therapies for sensory integration.

The review is interesting and well written; however, a few additions could enhance its overall completeness and depth.

In Section 3, which addresses key neuroplasticity disruptions in ASD, the authors mention some pathways and genes linked to synaptic function. This section would benefit from the inclusion of additional key genes such as BDNF, MECP2,  SNAP-25, or others which play established roles in synaptic function and are strongly implicated in autism. Including these genes would provide a more comprehensive molecular perspective.

Furthermore, in Section 4, the authors provide a broad and insightful overview of sensory dysregulation in children with ASD, offering a detailed explanation of neuroanatomical and neurotransmitter abnormalities associated with different aspects of sensory processing.

However, in Section 5, when discussing conventional and emerging therapies, the analysis is mostly limited to their symptom-reducing effects. A more in-depth exploration of how these interventions may act on the underlying neurobiological mechanisms would enhance the scientific value of this section and offer a clearer picture of the potential therapeutic impact.

Finally, the review does not address the role of parental involvement in therapeutic interventions.

Author Response

Dear Reviewer,

We sincerely thank you for the thoughtful and constructive comments, which have greatly helped improve the quality and scientific depth of our manuscript. Below, we provide a point-by-point response to each suggestion, along with an explanation of the changes made in the revised version.

  1. Comment 1: “In Section 3, which addresses key neuroplasticity disruptions in ASD, the authors mention some pathways and genes linked to synaptic function. This section would benefit from the inclusion of additional key genes such as BDNF, MECP2, SNAP-25, or others which play established roles in synaptic function and are strongly implicated in autism.”
    Response: Thank you for this valuable suggestion. In response, we have expanded Section 3 to include a more detailed discussion of additional genes relevant to synaptic plasticity in ASD. Specifically, we have added information on BDNF, MECP2, SNAP-25. We believe this addition significantly enhances the molecular perspective of our review (see revised Section 3, lines 177–199).
  2. Comment 2: “Furthermore, in Section 4, the authors provide a broad and insightful overview of sensory dysregulation in children with ASD, offering a detailed explanation of neuroanatomical and neurotransmitter abnormalities associated with different aspects of sensory processing. However, in Section 5, when discussing conventional and emerging therapies, the analysis is mostly limited to their symptom-reducing effects. A more in-depth exploration of how these interventions may act on the underlying neurobiological mechanisms would enhance the scientific value of this section and offer a clearer picture of the potential therapeutic impact.”
    Response: We fully agree, and thank you for highlighting this point. We have carefully revised Section 5 to include a more comprehensive analysis of the mechanistic basis of each therapy. These modifications provide a clearer picture of how therapies may influence the underlying neurobiology of ASD (see revised Section 5 and 6).
  3. Comment 3: “Finally, the review does not address the role of parental involvement in therapeutic interventions.”
    Response: Thank you for this excellent observation. In response, we have added a new paragraph in Section 5 discussing the importance of parental involvement in therapeutic interventions. (see new paragraph in Section 5, lines 488–507).

Once again, we thank you for your helpful feedback. We hope the revised manuscript is now significantly strengthened and more informative for readers.

Sincerely,

The Authors

Round 2

Reviewer 1 Report

Comments and Suggestions for Authors

No comments. 

Author Response

Dear Reviewer,

We would like to sincerely thank you for taking the time to read our manuscript and for the positive evaluation. Although no specific comments were raised, we have made additional improvements to the manuscript.

In particular, we have:

  • Carefully revised the language throughout the text to enhance clarity and readability;
  • Reviewed and corrected the reference list, adding missing citations and ensuring accuracy and consistency in formatting.

We are grateful for your contribution to the review process, which helped us to further improve the quality of our work.

Sincerely,

The Authors

Reviewer 2 Report

Comments and Suggestions for Authors

The authors have addressed all the concerns, and the manuscript is now greatly improved.
Minor revisions suggested include:

  1. A careful check of the cited references. In particular, regarding the discussion of SNP25, there are currently no bibliographic sources cited to support the statements made.

  2. A review of the writing style, as several newly added paragraphs begin in a repetitive manner, often starting with "This" or "These".

Author Response

Dear Reviewer,

We would like to sincerely thank you for the positive evaluation and constructive suggestions that helped us improve the quality of the manuscript.

Comment 1:
A careful check of the cited references. In particular, regarding the discussion of SNP25, there are currently no bibliographic sources cited to support the statements made.

Response:
Thank you for pointing this out. We have carefully reviewed the section discussing SNAP25 and have added appropriate references to support the statements made. Furthermore, we reviewed and corrected the references, adding missing citations and adding more appropriate ones.

Comment 2:
A review of the writing style, as several newly added paragraphs begin in a repetitive manner, often starting with "This" or "These".

Response:
We appreciate this stylistic observation. We have carefully revised the manuscript, particularly the newly added paragraphs, to improve sentence variety and eliminate repetitive sentence openings. We have restructured the affected sentences to enhance the overall flow and readability of the text.

Once again, we thank you for your valuable feedback and for acknowledging the improvements made to the manuscript.

Sincerely,

The Authors